# Study on Dissociation and Chemical Structural Characteristics of Areca Nut Husk

**DOI:** 10.3390/molecules28031513

**Published:** 2023-02-03

**Authors:** Jianbo Yuan, Haonan Zhang, Hui Zhao, Hao Ren, Huamin Zhai

**Affiliations:** Jiangsu Co-Innovation Center for Efficient Processing and Utilization of Forest Resources, Jiangsu Provincial Key Lab of Pulp and Paper Science and Technology, Nanjing Forestry University, Nanjing 210037, China

**Keywords:** lignocresol, lignin, phase separation system, fiber morphology

## Abstract

From the perspective of full-component utilization of woody fiber biomass resources, areca nut husk is an excellent woody fiber biomass feedstock because of its fast regeneration, significant regeneration ability, sustainability, low cost, and easy availability. In this study, fiber cell morphologies, chemical compositions, lignin structures, and carbohydrate contents of areca nut husks were analyzed and compared with those of rice straw, and the application potentials of these two materials as biomass resources were compared. We found that areca nut husk fibers were shorter and wider than those of rice straw; areca nut husk contained more lignin and less ash, as well as less holocellulose than rice straw; areca nut husk and rice straw lignin were obtained by ball milling and phase separation, and areca nut husk lignin was found to be a typical GHS-type lignin. Herein, the yield of lignocresol was higher than that of milled wood lignin for both raw materials, and the molecular size was more homogeneous. Tricin structural monomers were discovered in the lignin of areca nut husk, similar to those present in other types of herbaceous plants. Structures of areca nut husk MWL (AHMWL) and AHLC were comprehensively characterized by quantitative NMR techniques (that is, ^1^H NMR, ^31^P NMR, and 2D NMR). The molecular structure of AHLC was found to be closer to the linear structure with more functional groups exposed on the molecular surface, and the hydroxyl-rich *p*-cresol grafting structure was successfully introduced into the lignin structure. In addition, the carbohydrate content in the aqueous layer of the phase separation system was close to the carbohydrate content in the raw material, indicating that the phase separation method can precisely separate lignin from carbohydrates. These experimental results indicate that the phase separation method as a method for lignin utilization and structure study has outstanding advantages in lignin structure regulation and yield, and areca nut husk lignin is suitable for application in the same phase separation systems as short-period herbs, such as rice straw and wheat grass, and has the advantages of low ash content and high lignification degree, which will provide guidance for the high-value utilization of areca nut husk in the future.

## 1. Introduction

Areca nut (*Areca catechu* L.), an evergreen tree of the Palmaceae family, is one of the four southern Chinese herbs (namely areca nut, sand nut, Yizhi, and bahaldi) with erect stems, arboreal shapes, and oblong or ovoid fruits. Areca nut is primarily produced in south, southeast, and central Asian countries including India, Malaysia, and Myanmar [1,2]. It is mainly distributed in tropical areas, such as Hainan, Taiwan Province, and Yunnan, China. Moreover, it is widely cultivated in tropical Asia. According to the data released by the United Nations Food and Agriculture Organization, the total world output of areca nut in 2020 was 1,796,300 tons, with a harvest area of 1,226,100 hectares, of which, Hainan areca nut production ranks second in the world. According to the Hainan Bureau of Statistics 2020 statistical yearbook data, in 2020, 88,500 ha of areca nuts were harvested from Hainan, China, with an annual production of 283,300 tons of areca nuts, the highest yield per hectare in the world. Areca nut is a cash crop, and 1500–2000 plants can be grown per hectare. Areca nut cultivars in China have high yields of approximately 30 kg per plant, and areca nut can be used as an important traditional Chinese medicine and food processing product [3]. The economic value of the areca nut is high, and in addition to nuts, stalks and husks of this plant quickly grow and regenerate a huge amount of biomass as a renewable resource every year [4].

At present, the areca nut processing industry in China is underdeveloped, the products are single, and the grades of the products are not high, leading to the insufficient development potential of the areca nut industry in China. There are two main products of areca nut processing in China: dried areca nuts and jade, which are basically limited to the traditional consumption market. The mature husk of the areca nut accounts for 60–80% of the mass and volume of the whole fresh areca nut [5]. Because of the increase in areca nut output, as many as 1,077,800 tons of by-products of areca nut processing are obtained per year. Once the nuts are separated, numerous areca nut husks are discarded. Areca nut husk is primarily composed of cellulose (35–64.8%), hemicellulose, lignin (13–26%), and pectin (7%) [6]. It is a wood fiber resource material with high application potential, substantial regeneration ability, and high regeneration speed. However, the main treatment method for areca nut husk in many areca nut processing plants is to directly burn or bury this husk as waste, which results in the waste of biomass energy with potential application value. Moreover, areca nut husk does not rapidly decompose in nature, and rotten areca nut husk produces an unpleasant odor and causes environmental pollution.

From the perspective of full-component utilization of biomass resources, areca nut husk is an excellent lignocellulosic biomass raw material because of its fast regeneration, considerable regeneration ability, sustainable development, low cost, and easy availability. Therefore, exploring its application potential as a wood fiber resource is of significant importance to not only optimize the industrial structure of areca nut, promote the comprehensive utilization of areca nut as a cash crop, and improve the economic benefits of the areca nut industry, but also to reduce environmental pollution and slow down the exhaustion of petroleum resources. To date, some preliminary studies have been reported on the high value-added application potentials of areca nut husk [7,8,9]. Some of these studies were focused on the analysis and utilization of areca nut husk extracts [10], some paid attention to the fibers in areca nut husks and their application in fiber-reinforced composites [11], some used areca nut husks as raw materials to prepare nanocellulose [12], some employed these husks as fillings of foaming materials [13], and some employed areca nut husks as a metal-free cathode of microbial fuel cell [14]. Although these studies have different emphases, they have not comprehensively analyzed the fiber cell morphology, the contents of the main chemical components of the cell wall, the microchemical structure, and its potential for full-component utilization of lignocellulose biomass resources from the perspective of a renewable lignocellulose resource, which has limitations.

Rice straw, as agricultural waste with a huge yield, has so far been studied by researchers for its chemical composition and chemical structural characteristics [15,16] and has the potential for high value utilization in the process of biomass refining [17]. Through the fundamental research methods of wood chemistry and fiber morphology, herein, fiber morphology, basic chemical composition, lignin structure [18,19], and carbohydrate content and its composition of areca nut husk were systematically examined and compared with those of rice straw (typical straw of Gramineae); additionally, the application potentials of these two short-period agricultural and forestry waste wood fibers as biomass resources were compared. The results of this study will provide useful and reliable basic data for the areca nut husk food processing industry chain and an important reference for improving the economic benefits of the areca nut industry.

## 2. Results and Discussion

### 2.1. Analyses of the Chemical Compositions of Raw Materials

The chemical compositions of the raw materials (namely, areca nut husk and rice straw) used in this study are presented in Table 1. Areca nut husk had a lignin content of 28.75%, which was significantly higher than that of rice straw (16.40%); this was consistent with the higher strength of areca nut husk than that of rice straw—that is, the degree of lignification of areca nut husk was higher than that of rice straw. Holocellulose contents of areca nut husk and rice straw were 56.10 and 62.55%, respectively, and the sums of the contents of lignin and holocellulose were 84.85 and 78.95% for areca nut husk and rice straw, respectively; the benzene-alcohol extractive contents of areca nut husk and rice straw were 10.23 and 5.00%, respectively. This indicates that the content of compounds such as waxes, tannins, and lipids in areca nut husk was more than that of rice straw, which is macroscopically manifested by the smooth surface of areca nut husk. The sums of the contents of lignin, holocellulose, and benzene-alcohol extractives for areca nut husk and rice straw were 95.08 and 83.95%, respectively, and the remaining residues were mainly ash. Areca nut husk is a typical lignocellulosic resource in terms of its composition, and its ash content is considerably lower than that of rice straw because of its fast regeneration rate. Consequently, areca nut husk is a characteristic lignocellulosic resource with a certain compositional superiority. Additionally, the hot water extractives of areca nut husk and rice straw were 21.33 and 23.70%, respectively, and the 1% NaOH extractives were 40.92 and 52.00%, respectively. These extract contents imply that areca nut husk and rice straw have some similarities as their cell walls are loosely organized, and hemicellulose-like polysaccharides, etc., are easily soluble in hot water and alkaline solutions.

### 2.2. Investigation of the Fiber Morphologies of Raw Materials

Fiber morphologies of areca nut husk and rice straw after dissociation were observed using a light microscope (Figure 1). Areca nut husk and rice straw contained several miscellaneous cells (Figure 1), which primarily included finely shredded thin-walled cells, and serrated epidermal cells of rice straw were also observed (Figure 1b). Comparison of the morphologies of areca nut husk (Figure 1a) and rice straw (Figure 1b) implied that the number of heterocytes in areca nut husk was more than that in rice straw. The fiber cells of rice straw exhibited more elongated characteristics, whereas those of areca nut husk were relatively short and thick. Lengths, widths, aspect ratios, cell lumen diameters, cell wall thicknesses, wall lumen ratios, and curls of fiber cells of both raw materials were analyzed by a fiber morphology analyzer and Nano Measure, and the results are provided in Table 2. The average length (0.77 mm), average width (12.59 μm), and aspect ratio (61.16) of areca nut husk fiber were smaller, higher, and lower than those of rice straw (1.01 mm, 8.07 μm, and 125.15), respectively. This result indicates that although the aspect ratio of areca nut husk fiber is lower than that of rice straw, it meets the aspect ratio requirement (>45) of the pulp and paper industry [20]. Thus, areca nut husk has potential for application in the syntheses of low-strength papers.

Additionally, the softness of a fiber can be characterized by the degree of crimp of the fiber and the wall/cavity ratio [21]. Generally, the high degree of crimp of the fiber and the small wall-cavity ratio, indicating good flexibility of the fibers and strong inter-fiber bond. In the process of pulp and paper making, it will make the bond strength of the paper high. The degrees of crimp and wall-cavity ratios of areca nut husk and straw fibers were 3.3% and 1.13 and 4.4% and 0.92, respectively (Table 2). The wall-cavity ratio of areca nut husk is larger than that of straw, and the degree of crimp is smaller than that of straw, so in terms of fiber softness, areca nut husk does not have the advantage, and these data provide important reference data for how to use areca nut husk on the fiber scale.

### 2.3. Milled Wood Lignin (MWL) and Lignocresol (LC) Isolated from Areca Nut Husk and Rice Straw

#### 2.3.1. Yield of Lignin Isolated from Two Raw Materials

Yields of MWL and LC samples isolated from areca nut husk and rice straw are presented in Table 3. The yields of areca nut husk MWL (AHMWL) and areca nut husk LC (AHLC) were 6.80 and 11.82% relative to those of defatted wood flour and 25.92 and 94.07% relative to those of Klason lignin, respectively (Table 3). The yields of rice straw MLW (RSMWL) and rice straw LC (RSLC) were 1.30 and 4.63% relative to those of defatted straw flour and 21.48 and 87.59% relative to those of Klason lignin, respectively. Since MWL is the most representative natural lignin among the existing isolated lignins, this study intends to confirm the basic structural characteristics of lignin in areca nut husk, which regenerate very quickly, by isolating areca nut husk MWL and comparing it with a typical Graminaceae resource, rice straw. Additionally, phase separation enables the separation of lignin in essentially full quantification. Based on the method of Funaoka [22,23], our group performed a series of methods to separate lignins from different lignocellulosic resources and modify their structures and discovered that using a phase separation system, softwood kraft lignins could be extracted from herbaceous bamboo [24], woody horsetail pine [25], poplar [26], and pulp black liquor as lignin products with normalized structures, controlled benzyl position of activity, tunable phenolic hydroxyl contents, and condensed structures. Therefore, herein, phase separation systems were prepared using MWLs, and LCs extracted from areca nut husk and rice straw, and the application potentials of MWLs and LCs as lignocellulosic resources were evaluated by comprehensively comparing the changes in the structural characteristics of lignins isolated from areca nut husk and rice straw after their conversion to LCs. The yield of LC was approximately 90% or higher, which was significantly higher than that of ground lignin (approximately 20%); this indicated that in the phase separation system, the areca nut husks of herbaceous woody resources can separate lignin from carbohydrates almost quantitatively in the two-phase solvent system of acid and organic solvent as the coniferous and broad-leaved woods of woody resources. Because of the high isolation yield of LC when compared with that of MWL, subsequent studies of the structural characteristics, such as the linkage mode between structural units and the quantification of functional groups, of lignin will be more accurate and representative. Upon comparing the yields of AHLC and RSLC, it was found that the yield of AHLC was substantially higher than that of RSLC; this was attributed to the shorter growth periods and lower lignification degrees of rice straw plants, which are softer in appearance and have looser cell wall structures than those of perennial areca nut husk. Thus, in the phase separation system, quantitative separation of lignin and carbohydrates could be achieved for both areca nut husk and rice straw; however, because of the relatively low lignification degree of rice straw, more small molecules of LCs were lost in the subsequent multistep refining after the conversion of lignin to LC, leading to low final yield of RSLC. Therefore, the yield of AHLC is superior to those of the LCs of annual herbs such as rice straw.

#### 2.3.2. Fourier Transform Infrared (FTIR) Spectroscopy

FTIR spectra of AHMWL, AHLC, RSMWL, and RSLC are shown in Figure 2. The absorption peaks at 3432 cm^−1^ were ascribed to the alcoholic and phenolic hydroxyl groups of lignin, and the absorption peaks at 2929 and 2850 cm^−1^ originated from the C-H asymmetric and symmetric contraction vibrations of methyl and methylene groups, respectively, implying the presence of side-chain structural units in lignin. The spectra of both AHMWL and AHLC demonstrated peaks at 1607, 1510, and 1458 cm^−1^ corresponding to the skeletal vibrations of the lignin benzene ring, indicating that AHLC retained the basic structure of the lignin benzene ring during separation. The relatively small absorption peak at 1700 cm^−1^ was attributed to nonconjugated ketone and carbonyl stretching vibrations [27], and the absorption peak at 1655 cm^−1^ was ascribed to conjugated carbonyl stretching vibration. The absorption peak at 1362 cm^−1^ was attributed to the telescopic vibrations of phenolic hydroxyl groups [28], and the intensity of this absorption peak was relatively stronger for AHLC than that for AHMWL, which was due to the better accessibility of the phenolic hydroxyl groups of *p*-cresol in AHLC. A vibrational absorption peak of the benzene ring skeleton of the G-type lignin unit was observed at 1260 cm^−1^, and this peak was sharper for AHMWL and AHLC than that for RSMWL and RSLC, respectively, suggesting the existence of more methoxy groups in the guaiacyl (G)-type structural unit of areca nut husk [29]. The spectra of both RSMWL and AHMWL exhibited C-O-C stretching vibrational absorption peaks of the ester group of the p-hydroxyphenyl (H)-type lignin structural unit at 1164 cm^−1^ [30], indicating structural features typical of grass-like lignin. Furthermore, the C-O stretching vibration peak of the benzene ring of the syringyl (S)-type lignin unit was noticed at 1127 cm^−1^. The absorption peaks at 815 cm^−1^ in the spectra of both AHLC and RSLC were attributed to the variable angle vibrations of neighboring H atoms on the benzene ring of *p*-cresol. In summary, no significant structural differences were observed between AHMWL and RSMWL and AHLC and RSLC, and they all had typical GHS-type lignin structures.

### 2.4. Proton Nuclear Magnetic Resonance (^1^H NMR) Spectroscopy

^1^H NMR spectra of AHMWL, AHLC, RSMWL, and RSLC samples are depicted in Figure 3, and the results of the integration calculations are provided in Table 4. The methoxy contents of AHMWL and AHLC and RSMWL and RSLC were 19.59 and 7.94% and 17.48 and 11.13%, respectively. The lower methoxy contents of AHLC and RSLC as compared to those of AHMWL and RSMWL were ascribed to the absence of methoxy groups in the structures of the *p*-cresol moieties of AHLC and RSLC; the addition of *p*-cresol into lignin resulted in the introduction of a side chain with a benzene ring without a methoxy group at the active benzyl position of the lignin backbone structure, and, therefore, the average methoxy contents of AHLC and RSLC were lower. Based on the ^1^H NMR integral values, the contents of *p*-cresol in AHLC and RSLC were calculated to be 0.77 and 0.79 mol/C9, respectively; the relatively lower content of *p*-cresol in AHLC than that in RSLC was attributed to the higher lignification degree and more compact cell wall structure of areca nut husk than those of rice straw. Additionally, the intensities of the absorption peaks at 1.6–2.4 ppm corresponding to ^1^H of the methyl group were significantly higher for AHLC and RSLC as compared to those for AHMWL and RSMWL, implying successful introduction of *p*-cresol into the natural lignin structure [31].

### 2.5. Analyses of the Molecular Weights of Isolated Lignins

Number-average molecular weights (Mn) of AHMWL, AHLC, RSMWL, and RSLC were 5629, 3616, 4537, and 2348, respectively; the weight-average molecular weights (Mw) were 14,636, 5042, 9255, and 3239, respectively; and the polydispersity index (PDI (Mw/Mn) was 2.60, 1.39, 2.04, and 1.38, respectively (Table 5). The PDI values of LC samples were smaller than those of the MWL samples and were closer to 1. Molecular weight distributions and Mw values of the MWL samples were relatively wider and larger, respectively [32]. These results suggested that the active benzyl sites of lignin molecules were selectively cleaved under acidic conditions during phase separation to form Cα-C positive ions. As a large amount of *p*-cresol was present in the solution system, it was effectively incorporated into the electron-deficient sites of the Cα-C positive ions due to its small molecular size, the presence of numerous electron-donating groups in it, and rapidity, causing regular fragmentation of the lignin network-like macromolecules into linear macromolecules with lower molecular weights and relatively homogeneous molecular sizes. The abovementioned results indicated that owing to the higher lignification degree of areca nut husk than that of rice straw, the molecular weight of the lignin separated from areca nut husk by either physical or chemical methods was slightly larger than that of the lignin separated from rice straw; nevertheless, phase separation demonstrated an absolute advantage over the milled-wood method in terms of the molecular weight dispersion ratio. These data provide an important basis for the future industrialization and large-scale application of lignin.

### 2.6. Investigation of the ^31^P-NMR Spectra of AHMWL and AHLC

^31^P NMR spectra of AHMWL and AHLC are shown in Figure 4, and the contents of the functional groups of AHMWL and AHLC obtained based on the ^31^P NMR spectra and calculated according to the method shown in 2.10 are presented in Table 6. The phenolic hydroxyl content of AHLC was substantially higher than that of AHMWL due to the presence of a large amount of *p*-cresol in AHLC. Moreover, the *p*-hydroxyphenyl content of AHLC (2.03 mmol/g) was significantly higher than that of AHMWL (0.85 mmol/g), which was attributed to the introduced cresyl group having a *p*-hydroxyphenyl-type chemical structure. Additionally, ^31^P NMR spectra indicated that after phenolic modification, the aliphatic hydroxyl contents of lignin samples decreased (AHMWL: 2.77 mmol/g and AHLC: 1.24 mmol/g), which was ascribed to the grafting of *p*-cresol groups on the lignin side chain, replacing part of the aliphatic hydroxyl functional group, and the introduction of non-aliphatic hydroxyl *p*-cresol groups, decreasing the original aliphatic hydroxyl ratio, into the natural lignin structure [33].

### 2.7. Two-Dimensional (2D) NMR Spectroscopy

The main purpose of this study was to investigate the feasibility and potential of phenolization modification of areca nut husk lignin and changes in the structure of lignin after phenolization modification. Therefore, after the lignin samples were obtained by phase separation, 2D NMR spectroscopy of the MWL and LC samples was performed to examine the detailed structural changes of lignin. The signals in the 2D NMR spectra of AHMWL, AHLC, RSMWL, and RSLC were analyzed; the spectra of the lignin aliphatic region (δC/δH: 18–25/1.8–2.5), side-chain region (δC/δH: 50–90/2.5–6.0), aromatic region (δC/δH: 90–160/6.0–8.0), and the primary basic linkage structures and structural units are depicted in Figure 5, Figure 6 and Figure 7. The relative quantification method of Wen [34] was applied to evaluate the integration of each structural unit, and the relative fractions of the basic structural units and bond types are provided in Table 7. The 2D-heteronuclear single quantum coherence NMR attributions of MWL and LC are presented in Table 8. The aromatic ring region of the spectrum of AHMWL exhibited signals related to G (67.92%), S (26.41%), and H (5.67%) units with correlation signals of 11.98 for G/H and 0.39 for S/G; the aromatic ring region of the spectrum of RSMWL demonstrated signals related to G (53.89%), S (37.71%), and H (8.40%) units with correlation signals of 6.42 for G/H and 0.70 for S/G. The G/H ratio for areca nut husk was considerably higher than that for rice straw, whereas the number of H units was lower than that of rice straw. This was ascribed to the higher lignification degree and denser cell wall structure of areca nut husk than those of rice straw. These results indicated that both areca nut husk and rice straw had typical graminar G/S/H-type lignin characteristics. Both AHMWL and RSMWL had numerous β-O-4 ether bond (A) structures in the side-chain regions, which could be identified by the correlation signals of their α, β, and γ positions. The corresponding γ-position correlation signals δC/δH were 63.13/4.30 and 63.25/4.32 for areca nut husk and rice straw, respectively. The α-, β-, and γ-position correlation signals of the β-5 (IC) structure were also noticed in the spectrum, and the MWL β-5 contents were 16.95 and 8.24% for areca nut husk and rice straw, respectively. Signals corresponding to the β-β resinol (B) structure were also observed in the spectrum. Signals related to another structure of β-β tetrahydrofuran (B′) were also noticed owing to acetylation at the γ-position, and the corresponding α-position correlation signals δC/δH and β-β contents of MWL were 83.15/4.83 and 82.21/4.98 and 10.54 and 18.91% for areca nut husk and rice straw, respectively. In addition to the signals of the G, S, and H units, signals associated with the structures of *p*-hydroxycinnamyl alcohol (F), acetylated *p*-hydroxycinnamyl alcohol (F′), ferulic acid ester (FA), *p*-coumarate (*p*CA), and tricin (T) were observed in the spectra. Furthermore, herein, the existence of T structure in areca nut husk was discovered for the first time, and the signals of T’2,6, T6, T8, and T3 of AHMWL were obtained at δC/δH = 103.77/7.27, 98.61/6.24, 94.09/6.58, and 105.20/7.04 ppm, respectively. The T signal was also noticed at the corresponding position in the spectrum of the benzene ring region of RWMWL. T is an important flavonoid that is extensively present in less lignified gram lignin such as bamboo, wheatgrass, and rice straw [35]. By integrating the areas of the T signals in the benzene ring regions of AHMWL and RSMWL, the content of T in RSMWL was determined to be larger than that in AHMWL. It was speculated that the presence of the T structure might have some correlation with the biosynthesis of lignin, and the lower the lignification degree, the more the T content [36]. The existence of T, a monomer, in areca nut husk, is a crucial revelation for us to obtain both natural and synthetic T from areca nut husk.

2D NMR spectra of the aliphatic (δC/δH: 18–25/1.8–2.5), side-chain (δC/δH: 50–90/2.5–6.0), and aromatic (δC/δH: 90–160/6.0–8.0) regions of AHLC and RSLC are depicted in Figure 7, and their related signal attribution tables are presented in Table 8. Moreover, similar to the cases of the abovementioned MWL samples, the integration of the relative content of each structural unit and the calculated results are provided in Table 7. The spectrum of the aromatic ring region of AHLC also demonstrated correlation signals of G (65.85%), S (28.19%), and H (5.96%) units with G/H and S/G ratios of 11.05 and 0.42, respectively. Compared with the G/H ratio 11.90 and S/G ratio 0.39 of areca nut husk MWL, it was found that areca nut husk LC did not undergo essential changes in structural unit composition. The spectrum of the aromatic ring region of RSLC exhibited the signals of G (52.71%), S (39.18%), and H (8.78%) units, with G/H and S/G ratios 6.00 and 0.74, respectively. Upon comparing the G/H and S/G ratios of RSMWL and RSLC, it was found that the structural unit compositions of RSLC and RSMWL were not considerably different. In both cases, the signals of the basic bond structures, including β-O-4 ether bond (A), γ-position acetylated β-O-4 ether bond (A′), β-5 (IC), and β-β resin alcohol (B), were observed in the spectra of the side-chain regions. However, due to the grafting of *p*-cresol to the Cα position of the propane side chain, the signals of all the bond types, except for those of β-5 (IC), were noticed owing to the higher spatial site resistance of β-5 (IC), which was attributed to the absence of an H atom at the Cα position after the introduction of *p*-cresol into the benzyl position. Therefore, the usual method of quantifying lignin unit linkage bonds by integrating the Cα-H signal was no longer applicable. In our previous study [31], we discovered appropriate agreement between the ratios of the linkage bonds by integrating the Cγ-H signal peaks of β-O-4, β-5, and β-β in MWL and comparing them with the Cα-H signal peaks. The contents of β-O-4, β-5, and β-β in AHLC and AHMWL were 74.72, 14.92, and 9.37% and 72.51, 16.95, and 10.54%, respectively. Furthermore, the comparison revealed that AHLC basically retained the basic bond-type composition of AHMWL, and its spectrum exhibited the relevant signals of F, F′, FA, *p*CA, and T. Signals corresponding to numerous xylan (X) structures were noticed in the spectrum of the side-chain region of AHMWL as compared to the case of AHLC, for which signals related to the X structures were rare and almost absent in the spectrum of the side-chain region. This suggested that the LCC structure was cleaved under strong acidic conditions, and more sugars were removed during phase separation. This implied that phase separation not only phenolized lignin but also purified it. Upon comparing the 2D spectra of AHMWL and AHLC with the experimental results, it was observed that AHLC retained the structural units and bonding ratio composition similar to MWL, which is the closest to the original lignin. In the spectrum of the aromatic ring region, the signals L2,6 (δC/δH: 129.69/7.09) and L6 (δC/δH: 108.3/6.72), attributed to AHLC, appeared as a new peak at δC/δH: 22.21/2.17 in the fatty region of AHLC as compared to the case of AHMWL; this was caused by the incorporation of *p*-cresol into the lignin structure and the presence of a large group of lignins, which led to the shift of these signals as compared to the original methyl signal (δC/δH: 20.17/2.17). These results demonstrated that the hydroxyl-rich *p*-cresol graft structure was successfully introduced into the lignin sample isolated by phase separation while maintaining the skeletal structure of lignin.

### 2.8. Molecular Structure Models of AHMWL and AHLC

Based on the FTIR, gel permeation chromatography, ^1^H NMR, ^31^P NMR, and 2D NMR results of spherical AHMWL and AHLC, the corresponding molecular structures were established using Chem3D 15.1. The corresponding schematics are shown in Figure 8 and Figure 9. As shown in Figure 8, the molecular structure of AHMWL exhibited some agglomeration, and many reactive functional groups were wrapped inside the molecule. Nevertheless, owing to the incorporation of *p*-cresol into the lignin structure, the molecular structure of AHLC was based on the selective cleavage of the benzyl position of natural lignin and the introduction of *p*-cresol at the Cα position. Therefore, the molecular structure of AHLC was regularly cut off, and the macromolecular model was closer to linearity, allowing the exposure of more functional groups on the molecular surface, which would be beneficial for subsequent studies on the application of AHLC. Additionally, the content of aliphatic hydroxyl groups in AHLC was slightly lower than that in natural lignin due to the grafting of *p*-cresol groups on the lignin side chain, which was also demonstrated by the quantitative ^31^P NMR spectrum. This phenomenon was also reflected in the corresponding molecular structure model.

### 2.9. Contents of the Carbohydrates Dissolved in the Aqueous Phase

Carbohydrate contents of areca nut husk and rice straw before phase separation are presented in Table 1, and the contents of the carbohydrates dissolved in the aqueous phase after phase separation are provided in Table 9. The contents of various monosaccharides before and after phase separations of areca nut husk and rice straw are depicted in Figure 10 and Figure 11, and almost all sugars were present in the aqueous phase after phase separation. The total contents of carbohydrates in areca nut husk and rice straw before phase separation were 56.03 and 62.41%, respectively, whereas the total contents of the carbohydrates dissolved in the aqueous phases after phase separations of areca nut husk and rice straw were 54.51 and 59.98%, respectively. Phase separation not only can effectively separate carbohydrates from LC and realize the quantitative transformation of lignin, but can also efficiently transfer the carbohydrate components quantitatively from the plant cell wall to the aqueous phase; furthermore, the sugars dissolved in the aqueous phase can be subsequently used for fermentation and other carbohydrate utilization studies [37]. These basic data provide a reliable experimental basis for the full-component utilization of the biomass of areca nut husk.

## 3. Materials and Methods

### 3.1. Materials

The raw materials used in this study were areca nut husks and rice straws, which were divided into 40 mesh and 60 mesh wood flours after crushing and sieving. After benzene-alcohol extraction for 48 h, the volatile solvent was dried by air and then used to prepare milled wood lignin (MWL) and lignocresol (LC). Chemicals used were: benzene, ethanol, sodium chlorite, glacial acetic acid, hydrogen peroxide, sulfuric acid, sodium hydroxide, acetone, ether, phosphorus pentoxide, pyridine, 1,4-dioxane, dichloroethane, and acetic anhydride and are all purchased from Nanjing Chemical Reagents Co., Ltd. (Nanjing, China); sodium hypochlorite was purchased from Zhenjiang Jiuyi Chemical Reagent Co., Ltd. (Zhenjiang, China); standard samples of deuterated chloroform, tetrahydrofuran, and polystyrene were purchased from Sigma-Aldridge (Shanghai) Trading Co., Ltd. (Shanghai, China); and DMSO-d6, cyclohexanol, and chromium acetoacetate were purchased from Shanghai Aladdin Biochemistry Technology Co., Ltd. (Shanghai, China).

Research manuscripts reporting large datasets that are deposited in a publicly available database should specify where the data have been deposited and provide the relevant accession numbers. If the accession numbers have not yet been obtained at the time of submission, please state that they will be provided during review. They must be provided prior to publication.

Interventionary studies involving animals or humans, and other studies that require ethical approval, must list the authority that provided approval and the corresponding ethical approval code.

### 3.2. Chemical Composition Analysis of Raw Materials

Detailed chemical composition analysis of areca nut husks and rice straws were carried out according to GB/T standard measurement methods. The holocellulose (GB/T2677.10-1995), klason lignin and acid-insoluble lignin (GB/T2677.8-1994), benzyl alcohol extracts (GB/T2677.6-1994), and ash content (GB/T2677.3-1993) of the raw materials were determined.

The monosaccharide composition was determined using gas chromatography (GC2010plus, Shimadzu enterprise management Co., Ltd., Shanghai, China) and the basis of the method presented in the literature [38]. Gas chromatographic conditions were as follows: SH-Rtx-1701 capillary column (14% cyanopropylphenyl/86% dimethyl polysiloxane: 30 m × 0.32 mm, 0.1 μm, Shimadzu, Kyoto, Japan). Split sampling was performed with an injection volume of 1 µL and a split ratio of 75:1. The oven temperature was set to 160 °C for 55 min. The heater temperatures of the injector and detector were maintained at 230 °C and 250 °C, respectively. The carrier gas was nitrogen (1.98 mL/min). The results are reported as relative peak areas. Xylose, mannose, galactose, arabinose, and glucose were used as monosaccharide standards.

### 3.3. Fiber Morphology Analysis

The two raw materials were cut into matchstick size and boiled in water several times until the samples sank to the bottom. The samples were submerged in a mixture of hydrogen peroxide and glacial acetic acid in a volume ratio of 1:1 and reacted at 60 °C until the samples turned white. The boiled samples were washed several times with water until they were not acidic when tested with pH paper. The fibers were squeezed as dry as possible, taken partially into a centrifuge tube, stained with a solution of 1% Fancy Red by mass for two minutes, and rinsed with distilled water to remove the floating color. Distilled water was added to the tube to make the fibers evenly dispersed, and then an appropriate amount of the fiber mixture was aspirated with a dropper and placed on the slide. The slide was covered, and filter paper was used to absorb excess water to make a temporary slide for observing fiber morphology by light microscope.

Afterwards, the length, width, wall thickness, and wall cavity values of the fiber micrographs of the raw material were determined using Nano Measurer 1.2 software.

### 3.4. Benzene-Alcohol Extractive of Raw Materials

The dried areca nut husks and rice straws were extracted with a mixed solution of benzyl-alcohol (2:1, *v*/*v*) at 80 °C for 48 h. The dewaxed raw material was obtained after removing the extract and then dried in a ventilated place. The dried wood powder was then powdered with a micro plant crusher and used as a subsequent preparation of milled wood lignin and lignocresol.

### 3.5. Separation and Purification of Milled Wood Lignin

Ball milling conditions: 2.5 g of defatted wood powder was placed in an 80 mL zirconia jar containing 25 ZrO2 balls (1 cm diameter), and the ball milling frequency was 500 rpm. A planetary ball mill (PILVERISETTE7, PM, Germany) was used, with 5 min of ball milling and 5 min of intermittent grinding, avoiding temperatures above 40 °C, and accumulating 24 h of ball milling.

Then 20 g of milled dewaxed wood powder was stirred magnetically with 200 mL of 96% dioxane (purified by NaOH) for 2 h at room temperature and centrifuged to collect the supernatant. The precipitate was extracted again, and the extracted liquid was collected; the extracted liquid was concentrated drop by drop into an appropriate amount of 95% ethanol and centrifuged to remove the hemicellulose precipitate, and the lignin liquid was concentrated again drop by drop into 10 times the volume of acid water (pH = 2.0, hydrochloric acid), centrifuged to collect the precipitate, and then freeze dried and vacuum dried to crude lignin. The crude lignin was dissolved into 90% acetic acid, and then the solution was added drop by drop to 10 times the volume of acid water, centrifuged and lyophilized to obtain the precipitate, which was fully dissolved in dichloroethane/ethanol; then it was added drop by drop to anhydrous ethyl ether in an ice water bath, centrifuged to collect the precipitate, washed by centrifugation with ethyl ether three times, and freeze dried to obtain the purified MWL. The specific experimental procedure is shown in Figure 12; the areca nut husk milled wood lignin (AHMWL) and rice straw milled wood lignin (RSMWL) were prepared according to this procedure.

### 3.6. Separation and Purification of Lignocresol

Next, 1 g (accurate to 0.0001) of dewaxed raw material wood powder was weighed in a 100 mL volumetric beaker, 10 mL of *p*-cresol was measured into the beaker, and 20 mL of 72% concentrated sulfuric acid was added and stirred after the wood powder was fully moistened. After the cell wall was fully swollen and placed on a magnetic stirrer for a total of 1 h, the reaction mixture (3500 rpm, 25 °C) was centrifuged; the upper organic phase was extracted drop by drop. The crude lignin phenol was obtained by centrifugation (3500 rpm, 5 °C) and was dropped into ether in an ice-water bath; after that, the crude lignin phenol was dissolved in acetone, stirred slowly at room temperature, and protected from light, and the acetone solution was centrifuged (3500 rpm, 5 °C). After the carbohydrates were dissolved, the collected acetone solution of lignin phenol was concentrated by spin evaporation and then added dropwise into ether and was centrifuged again at low temperature. To collect the insoluble flocculent precipitate in ether, it was collected again by centrifugation at low temperature to obtain the LC product. The LC samples were dried under vacuum at 40 °C for 24 h and micronized to obtain the LC samples for subsequent instrumental analysis. Areca nut husk lignocresol (AHLC) and rice straw lignocresol (RSLC) were prepared by this procedure.

### 3.7. Acetylation of Lignin

About 100 mg of MWL or LC sample was placed into a 5 mL reagent bottle with a cap, 1 mL of pyridine was added to dissolve the lignin, and it was dissolved fully; 1 mL of acetic anhydride solution was added and mixed well, and it was allowed to stand in a tinfoil wrapper for 48 h. An appropriate amount of distilled water was poured into a 50 mL conical flask, and the pyridine acetic anhydride mixture was placed into the conical flask drop by drop with a dropper after 48 h of reaction. A precipitate was produced after the drop, after which the precipitate was recovered from the mixture by centrifugation at 5000 rpm, 5 °C for 10 min, and this step was repeated three times. The precipitate was air dried, and the samples were vacuum dried to obtain acetylated milled wood lignin (AMWL) and acetylated lignocresol (ALC) samples.

### 3.8. FTIR Determination

Completely dried MWL and LC samples were mixed with dried and dewatered KBr in the ratio of 1:100 and the samples were prepared by the press method. The infrared spectra were measured with an infrared spectrometer (VERTEX 80V Bruker) with a resolution of 4 cm^−1^ and 32 scans in the wave number range 4000–500 cm^−1^.

### 3.9. ^1^H NMR Determination

The acetylated lignin sample was vacuum dried for 24 h. An amount of 20 mg of lignin sample and 3 mg of *p*-nitrobenzaldehyde internal standard were accurately weighed, added to a 1:3 mixture of deuterated pyridine and deuterated chloroform (0.15 mL of deuterated pyridine and 0.45 mL of deuterated chloroform), fully dissolved and transferred to a standard NMR tube, and detected with a Bruker AVANCE III 600 NMR instrument. The hydrogen spectra obtained were used to calculate the content of methoxy and the amount of *p*-cresol introduced according to the following method.

The formula for calculating the content of methoxy is as follows [39]:(1)$$\%OCH_{3}=28.28436-19.75004x$$
where *x* is the integral value of hydrogen in the aromatic nucleus in the range of 6.40–7.15 ppm, and the integral value of hydrogen in methoxy is in the range of 3.01–4.01 ppm.

Using the internal standard method, with *p*-nitrobenzaldehyde as the standard sample, the import amount of *p*-cresol is calculated as follows [37]:(2)$$I\left(wt\%\right)=\frac{OPwt}{151}\times \frac{4}{OPi}\times \frac{Ci}{Cn}\times \left(Cm-1\right)\times \frac{1}{Lwt}\times 100\%$$
(3)$$I\left(mol/C9\right)=\frac{Iwt\%}{Cm}\times \frac{Lm}{100-Iwt\%}$$
where *I*(*wt*%) is the percentage by mass of the imported *p*-cresol; *I*(*mol*/*C9*) is the moles of imported *p*-cresol per C9; *OPwt* is the mass of the internal standard (mg); *OPi* is the integral value of the hydrogen in the aromatic nucleus in the internal standard in the range of 7.95–8.26 ppm; *Ci* is the hydrogen in the methyl group in the imported *p*-cresol in the range of 1.60–2.40 ppm integral value; *Cn* is the number of hydrogen atoms in the methyl group in the imported *p*-cresol (=3); *Cm* is the relative molecular mass of the imported *p*-cresol108; *Lwt* is the mass of the *LC* weighed (mg); and *Lm* is the relative molecular mass of lignin unit.

### 3.10. ^31^P NMR Determination

The ^31^P-NMR analysis of AHMWL and AHLC was performed according to the literature [40]. About 20 mg of lignin sample was weighed and dissolved in 500 μL pyridine-d5/CDCl3 (1.6:1, *v*/*v*), followed by the addition of 100 μL of internal standard cyclohexanol (10.85 mg/mL, solvent pyridine-d5/CDCl3) and 100 μL of relaxant (5 mg/mL, solvent Pyridine-d5/CDCl3). An amount of 100 μL of phosphitylating agent (2-chloro-1,3,2-dioxaphospholane) was added into the mixed solvent to react for 10 min, and then the phosphorylated sample was transferred to a 5 mm NMR tube for phosphorus spectroscopy [41]. The peak of the internal standard cyclohexanol signal was at 145.1 ppm, which was used to calibrate the phosphorus spectrum signal.

The formula for calculating the functional group content is as follows:(4)$$x=\frac{1085A}{mM}$$
where: *x* is the functional group content (mmol/g), *m* is the mass (g) of the lignin sample that was weighed, *M* is the relative molecular mass of the internal standard (100.16 g/mol), and *A* is the integral value of the phosphorus spectrum of functional groups, which is equal to 149.8–145.4 ppm for aliphatic hydroxyl groups, 144.5–140.4 ppm for condensed phenolic hydroxyl groups, 140.4–138.2 ppm for noncondensed phenolic hydroxyl group of guaiacyl type structural unit (G), 138.2–137.2 ppm for noncondensed phenolic hydroxyl of *p*-hydroxyphenyl structural unit (H), and 135.8–134.0 ppm for carboxyl groups.

### 3.11. 2D NMR Determination

The MWL and LC samples of rice straws and areca nut husks were dried under vacuum and mixed at a ratio of 70 mg/0.7 mL of deuterated DMSO-d6 and transferred to a standard NMR tube through cotton filtration for detection using a Bruker AVANCEIII 600 NMR instrument from Bruker Biospin. The spectral widths of the ^1^H and ^13^C dimensions were 9615 Hz and 24,900 Hz, respectively. 2048 samples were taken in the 1 H dimension with a relaxation time of 1.5 s and 64 times of accumulation, and 256 samples were taken in the ^13^C dimension. 145 Hz was used for the hydrocarbon coupling constant, and zero punching was performed before Fourier transformation. The obtained 2D-HSQC NMR spectra were used for the structural analysis of MWL and LC.

A semiquantitative method [34] was used to calculate the unit composition and side-chain composition of lignin samples. The calculation formula of lignin unit composition is as follows:(5)$$I_{x}\%=\frac{I_{x}}{0.5IS_{2,6}+IG_{2}+0.5IH_{2,6}}$$
where: *IG*_2_, *IS*_2,6_, and *IH*_2,6_ are the integral values of position 2 of *G* unit, position 2,6 of *S* unit, and position 2,6 of *H* unit, respectively.

The calculation formula of lignin side-chain composition is as follows:(6)$$I_{y}\%=\frac{I_{y}}{I_{A}+I_{B}+I_{C}}$$
where: *I_A_*, *I_B_*, and *I_C_* are the integral values of β-O-4, β-β, and β-5 in α position, respectively, and are substituted into *I_y_* to obtain the percentage of different bond types relative to the total bond type *I_y_*%.

### 3.12. GPC Determination

The AHMWL, AHLC, RSMWL, and RSLC samples were dissolved in tetrahydrofuran solution and filtered through an organic syringe filter with a pore size of 0.22 μm. The molecular weight of lignin was determined by a Shimadzu high-performance gel permeation chromatographer (Shimadzu, Kyoto, Japan) equipped with an SPD-20A detector and an LC20AB pump. The chromatography was performed on Asahipak Shodex KF-804L (Showa Denko, Tokyo, Japan) polymer-based columns (300 mm, 8 mm ID). The injection volume was 25 μL, the flow rate was 1 mL/min, and the column temperature was 40 ℃. The calibration was performed using standard polystyrene for molecular weights of 46,300, 30,000, 20,000, 10,000, 4050, 2400, 1000, and 500.

### 3.13. Analysis of Aqueous Layer Sugars in Phase Separation Systems

An amount of 1.000 g of dewaxed areca nut husk or rice straw flour were weighed. Next, 10 mL of *p*-cresol and 20 mL of 72% concentrated sulfuric acid solution were added and stirred thoroughly for 1h. The upper organic phase was separated from the lower aqueous layer by centrifugation (3500 rpm, 25 °C, 15 min). The upper organic phase was removed, and 2 mL of the lower aqueous phase solution was aspirated into a blue-capped bottle and then was diluted with 74 mL of distilled water. The aqueous phase was hydrolyzed in a sterilizer at 121 °C for 1 h. The sugar content in the aqueous phase was determined using the GC method mentioned in Section 3.2 above.

## 4. Conclusions

Currently, areca nut husks are regarded as waste materials in the food processing industry; nevertheless, they have potential for application as an herbaceous woody fiber biomass feedstock. The cell wall structures of areca nut husks are relatively loose and finely fibrous; although the aspect ratio of areca nut husk (61) is smaller as compared to that of rice straw (125), it still meets the basic requirements for fiber formation in the pulp and paper industry. In terms of chemical composition, areca nut husk is composed of cellulose, hemicellulose, and lignin; contains more extracted substances than those of common graminaceous plants; and has a higher lignification degree than that of rice straw. Herein, lignin in areca nut husk was separated and extracted by ball milling and phase separation separately; moreover, rice straw was used as the control sample, and the structures of both AHMWL and AHLC, the natural lignin isolates from areca nut husks, were found to be similar to those of RSMWL and RSLC, respectively, which were formed by the typical three basic structural units G, S, and H. Because of their higher lignification degrees, both AHMWL and AHLC demonstrated higher molecular weights than those of RSMWL and RSLC, respectively, which makes them reflect the characteristics of larger molecular weight that makes them less likely to be lost with organic solvent resulting in higher yield in the crude lignin refining recovery stage. The phenolic hydroxyl content of AHLC modified by phase separation was as high as 3.72 mmol/g, and the aliphatic hydroxyl content of lignin was reduced to 1.24 mmol/g after phase separation. Additionally, T structural monomers (commonly found in graminaceous materials) were present in areca nut husk lignin. According to the cell wall structure of the areca nut husk tissue and structural characteristics of natural lignin, the *p*-cresol structure can be selectively introduced into the lignin molecule via phase separation, thus regulating the functional group distribution of the separated lignin product and its molecular weight PDI. Furthermore, the sugars in the aqueous phase of the phase separation system were quantified, and all the carbohydrate fraction in the cell wall was basically quantitatively converted into water-soluble sugars during separation and transformation. This study offers valuable basic data, a conducive raw material, and a pathway to improve the conversion efficiency of lignin for subsequent biomass refining using lignin-free plant polysaccharides as raw materials. In summary, areca nut husk, an herbal lignocellulosic resource, has application value and utilization potential at both fiber and molecular levels. The results of this study can provide an effective pathway and important reference for waste treatment in food development enterprises, forming a more complete industrial chain, achieving zero emissions, and fully realizing the added value of agricultural and forestry crops.

## Figures and Tables

**Figure 1 molecules-28-01513-f001:**
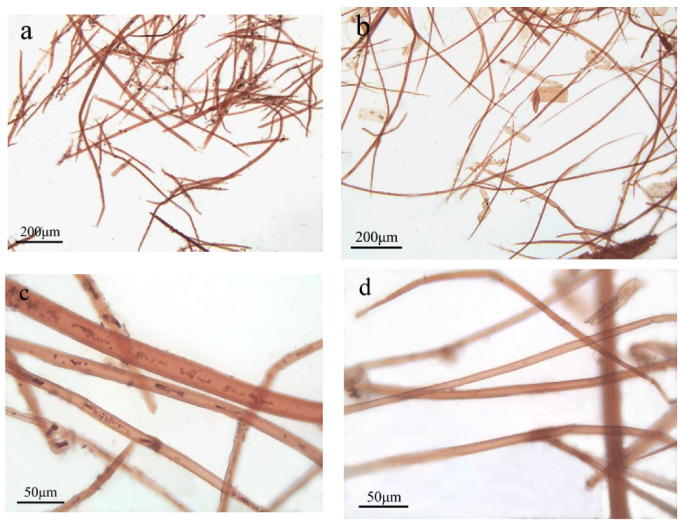
Morphological micrographs of areca nut husk and rice straw. (**a**) 100× micrograph of areca nut husk fiber morphology; (**b**) 100× micrograph of rice straw fiber morphology; (**c**) 400× micrograph of areca nut husk fiber morphology; (**d**) 400× micrograph of rice straw fiber morphology.

**Figure 2 molecules-28-01513-f002:**
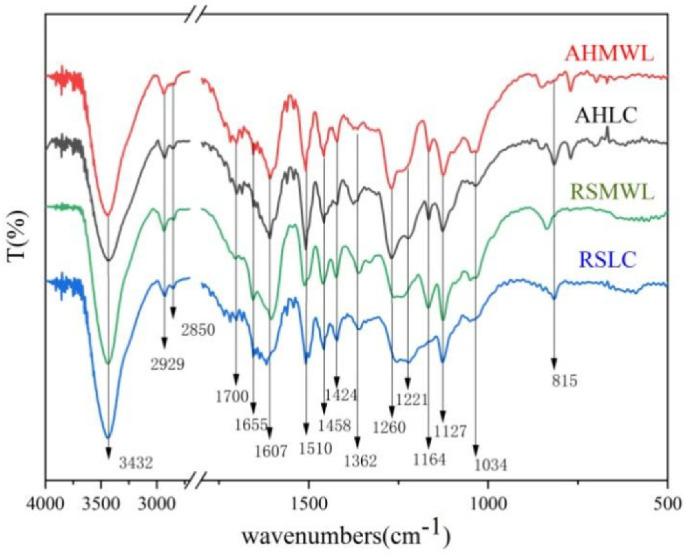
Infrared spectra of AHMWL, AHLC, RSMWL, and RSLC.

**Figure 3 molecules-28-01513-f003:**
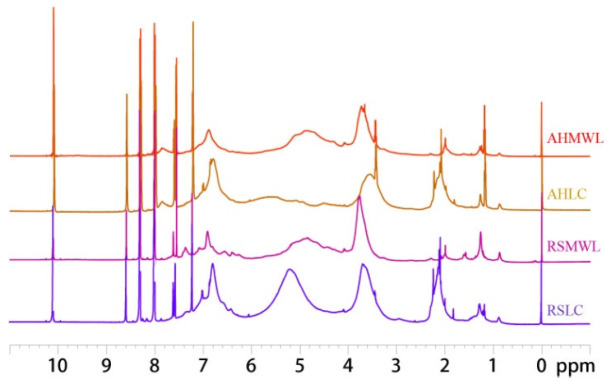
^1^H-NMR spectra of AHMWL, AHLC, RSMWL, and RSLC.

**Figure 4 molecules-28-01513-f004:**
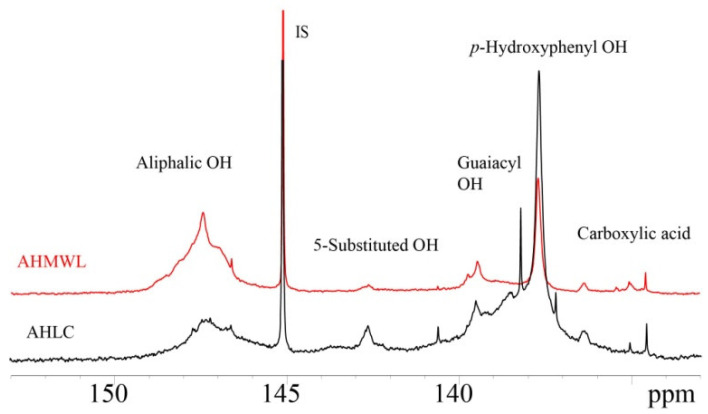
^31^P-NMR spectrum of AHMWL and AHLC.

**Figure 5 molecules-28-01513-f005:**
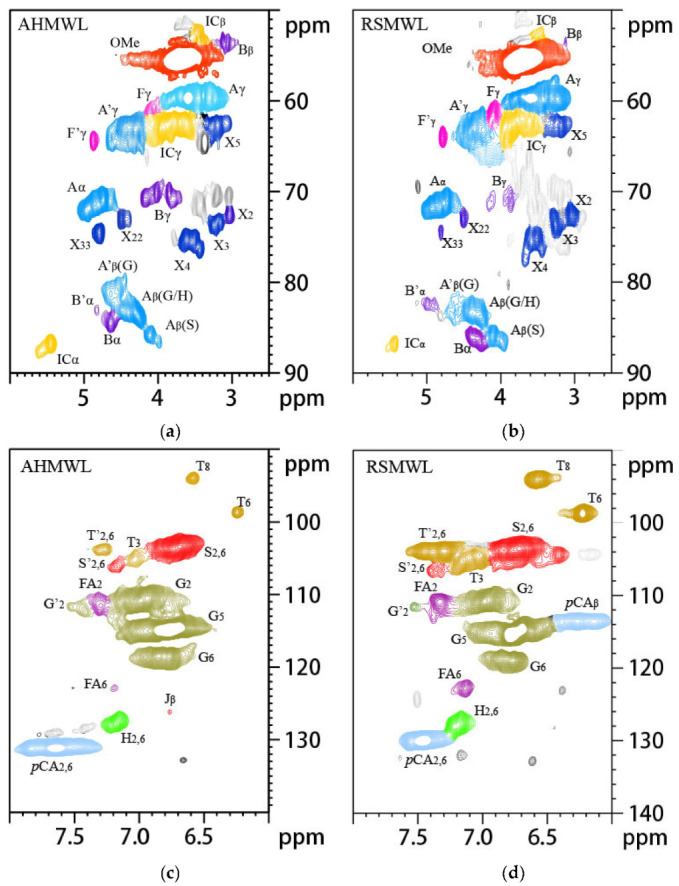
The 2D-HSQC NMR spectra of AHMWL and RSMWL: (**a**) The spectra of the AHMWL side-chain region; (**b**) The spectra of the RSMWL side-chain region; (**c**) The spectra of the AHMWL aromatic region; (**d**) The spectra of the RSMWL aromatic region.

**Figure 6 molecules-28-01513-f006:**
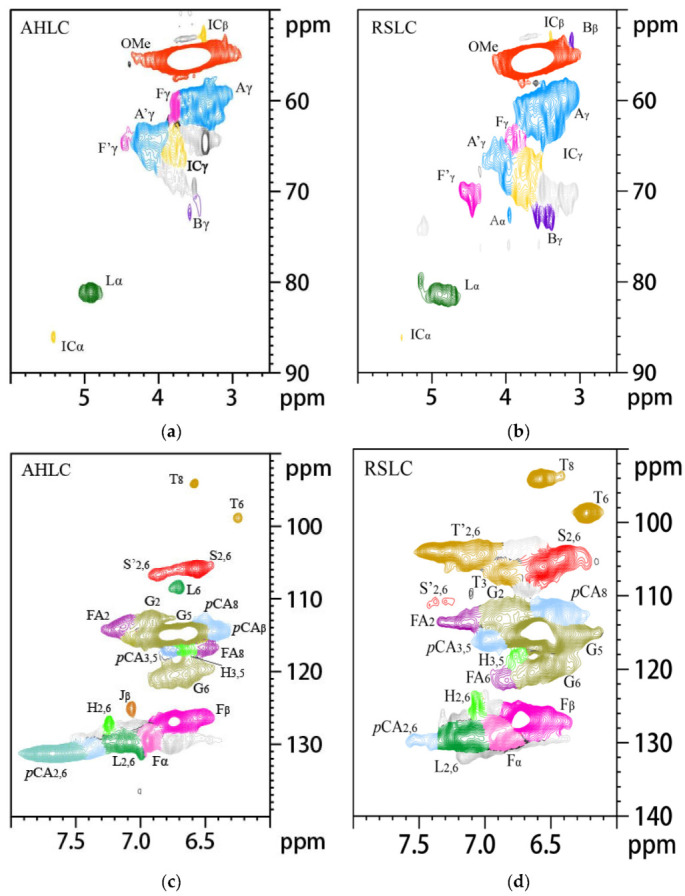
The 2D-HSQC NMR spectra of AHLC and RSLC: (**a**) The spectra of the AHLC side-chain region; (**b**) The spectra of the RSLC side-chain region; (**c**) The spectra of the AHLC aromatic region; (**d**) The spectra of the RSLC aromatic region; (**e**) The spectra of the AHLC aliphatic region; (**f**) The spectra of the RSLC aliphatic region.

**Figure 7 molecules-28-01513-f007:**
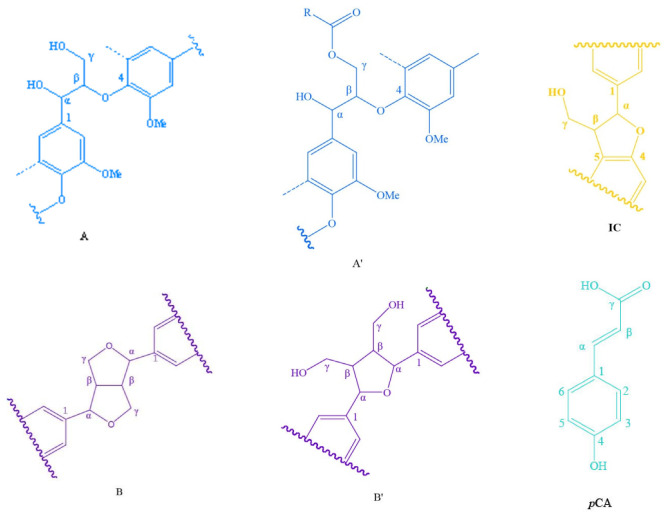
Main lignin unit and linkages in the lignin preparations: (**A**) β-O-4 alkyl-aryl ethers; (**A′**) β-O-4 alkyl-aryl ethers with acylated γ-OH with pcoumaric acid; (**B**) resinol structures formed by β-β/α-O-γ/γ-O-α linkages; (**B′**) tetrahydrofuran structures formed by β-β linkages; (**IC**) phenylcoumarane structures formed by β-5/α-O-4 linkages; (**F**) *p*-hydroxycinnamyl alcohol end-groups; (**F′**) *p*-hydroxycinnamyl alcohol end-groups acylated at the γ-OH; (***p*CA**) p-coumarates; (**FA**) ferulates; (**H**) phydroxyphenyl units; (**G**) guaiacyl units; (**G′**) oxidized guaiacyl unit linked a carbonyl at Cα; (**S**) syringyl units; (**S′**) oxidized syringyl units bearing a carbonyl at Cα; (**X**) xylopyranoside; (**L**) lignocresol; (**L′**) *p*-cresol; (**T**) tricin.

**Figure 8 molecules-28-01513-f008:**
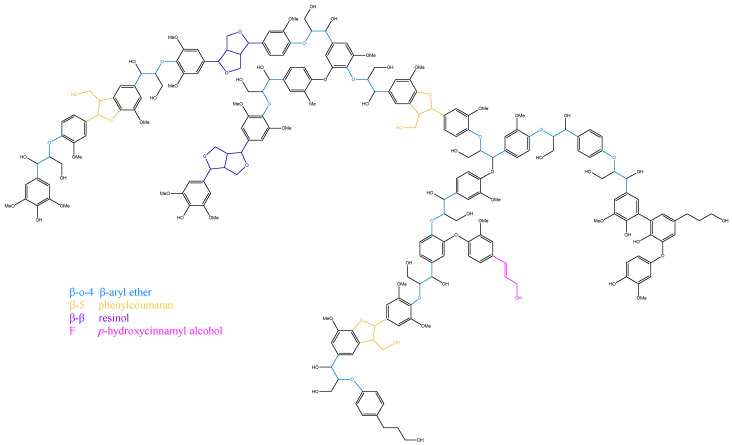
The molecular structure model of AHMWL.

**Figure 9 molecules-28-01513-f009:**
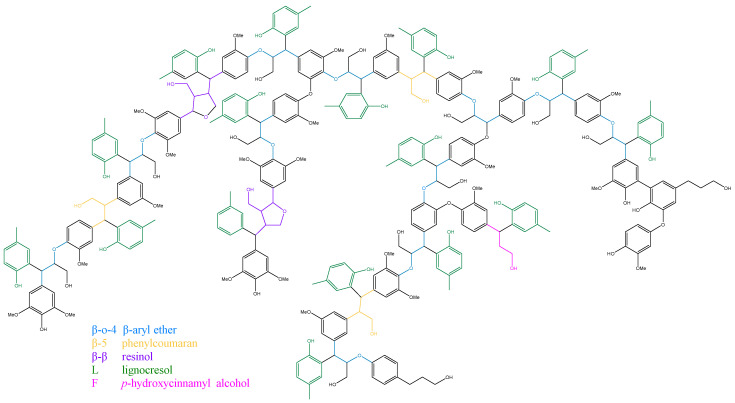
The molecular structure model of AHLC.

**Figure 10 molecules-28-01513-f010:**
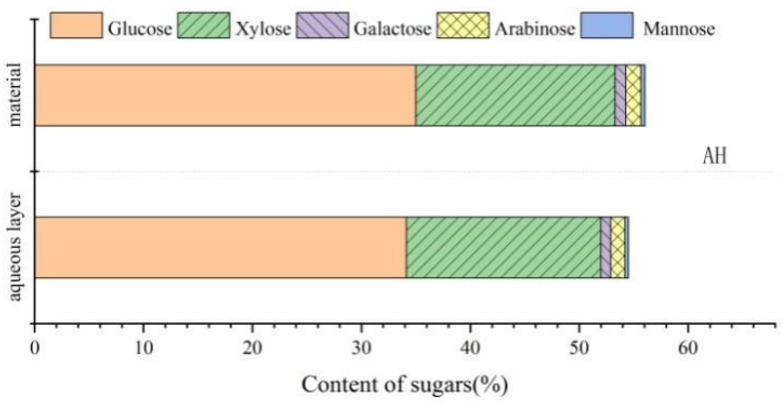
Analysis of carbohydrate content of areca nut husk before and after phase separation.

**Figure 11 molecules-28-01513-f011:**
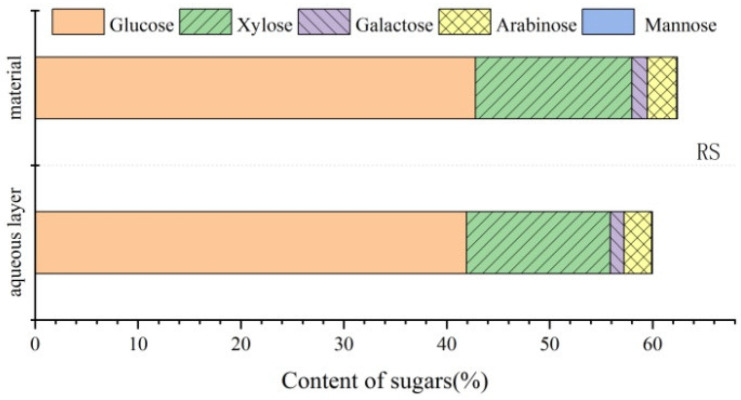
Analysis of carbohydrate content of rice straw before and after phase separation.

**Figure 12 molecules-28-01513-f012:**
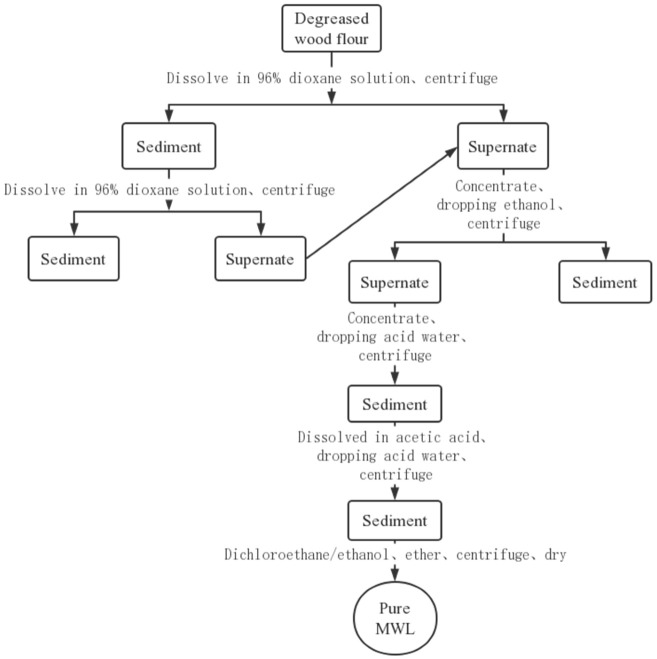
Flow chart of separation and purification of MWL.

**Table 1 molecules-28-01513-t001:** Compositional analysis of the AH and RS.

Chemical Composition	Areca Nut Husk	Rice Straw
Klason lignin (%)	26.56	13.30
Acid soluble lignin (%)	2.19	3.10
Total lignin content (%)	28.75	16.40
Holocellulose (%)	56.10	62.55
Glucose (%)	35.00	42.80
Xylose (%)	18.29	15.20
Galactose (%)	0.99	1.49
Arabinose (%)	1.41	2.84
Mannose (%)	0.34	0.08
Total carbohydrate content (%)	56.03	62.41
Benzene-alcohol extractive (%)	10.23	5.00
Hot water extractive (%)	21.33	23.70
1% NaOH extractive (%)	40.92	52.00
Ash (%)	3.53	14.31

**Table 2 molecules-28-01513-t002:** Fiber morphologies of areca nut husk and rice straw.

Raw Materials	Length /(mm)	Width /(μm)	Aspect Ratio	Cell Cavity Diameter /(μm)	Cell Wall Thickness /(μm)	Wall-Cavity Ratio	Degree of Crimp/%
AH	0.77	12.59	61.16	5.91	3.34	1.13	3.3
RS	1.01	8.07	125.15	4.20	1.94	0.92	4.4

**Table 3 molecules-28-01513-t003:** Yield of MWL and LC of two raw materials.

Lignin	Wood Flour Weight/g	Klason Lignin/g	Lignin Products/g	Yield (to Wood Flour %)	Yield (to Klason Lignin %)
AHMWL	3.97	1.05	0.27	6.80	25.92
AHLC	1.10	0.29	0.13	11.82	94.07
RSMWL	4.62	0.61	0.06	1.30	21.48
RSLC	1.08	0.14	0.05	4.63	87.59

**Table 4 molecules-28-01513-t004:** Methoxyl content and *p*-cresol introduction amount.

Sample	Methoxyl Content	*p*-Cresol Introduction Amount	
	wt%	wt%	mol/C9
AHMWL	19.59	-	-
AHLC	7.94	27.77	0.77
RSMWL	17.48	-	-
RSLC	11.13	30.49	0.79

**Table 5 molecules-28-01513-t005:** Molecular weight of AHMWL, AHLC, RSMWL, and RSLC.

Sample	Mn	Mw	PDI (Mw/Mn)
AHMWL	5629	14,636	2.60
AHLC	3616	5042	1.39
RSMWL	4537	9255	2.04
RSLC	2348	3239	1.38

**Table 6 molecules-28-01513-t006:** Functional group contents of AHMWL and AHLC.

Functional Group Contents (mmol/g)	AHMWL	AHLC
Aliphatic OH	2.77	1.24
5-Substiyuted OH	0.04	0.32
Guaiacyl OH	0.62	1.37
p-Hydroxyphenyl OH	0.85	2.03
Total phenolic OH	1.51	3.72
Carboxyl	0.02	0.23

**Table 7 molecules-28-01513-t007:** The units and linkages proportion of MWL and LC.

Sample	S/%	G/%	H/%	S/G	G/H	β-O-4/%	β-β/%
AHMWL	26.41	67.92	5.67	0.39	11.98	72.51	10.54
AHLC	28.19	65.85	5.96	0.42	11.05	74.72	9.37
RSMWL	37.71	53.89	8.40	0.70	6.42	72.85	18.91
RSLC	39.18	52.71	8.78	0.74	6.00	74.91	17.53

**Table 8 molecules-28-01513-t008:** The units and linkages proportion of MWL and LC.

Label	δC/δH (ppm) ^a^	δC/δH (ppm) ^b^	δC/δH (ppm) ^c^	δC/δH (ppm) ^d^	Assignment
L′-CH3	ND	20.17/2.17	ND	20.21/2.15	C−H in cresol-CH3 (L′)
L-CH3	ND	22.21/2.17	ND	22.81/2.18	C−H in lignocresol-CH3 (L)
ICβ	53.26/3.45	52.94/3.40	52.62/3.47	52.82/3.39	Cβ−Hβ in phenylcoumaran substructures (IC)
Bβ	53.79/3.06	ND	53.61/3.43	53.38/3.11	Cβ−Hβ in β−β (resinol) substructures (B)
OMe	55.47/3.73	55.46/3.58	55.6/3.73	55.67/3.69	C–H in methoxyls (OMe)
Aγ	59.78/3.60	61.09/3.44	59.78/3.60	60.46/3.46	Cγ−Hγ in β–O–4 substructure (A)
A′γ	63.13/4.30	64.18/4.25	63.12/4.36	67.11/3.97	Cγ−Hγ in γ-hydroxylated β–O–4 substructures (A′)
Fγ	61.17/4.11	61.10/3.77	61.98/4.11	64.93/3.88	Cγ−Hγ in *p*-hydroxycinnamyl alcohol (F)
F′γ	64.46/4.88	64.77/4.47	64.03/4.79	70.77/4.47	Cγ−Hγ in γ-acylated cinnamyl alcohol (F′)
ICγ	62.42/3.70	64.84/3.80	62.89/3.88	67.54/3.77	Cγ−Hγ in phenylcoumaran substructures (IC)
Bγ	70.34/3.84	ND	71.22/4.13	72.89/3.58	Cγ−Hγ in β−β (resinol) substructures (B)
Bγ	70.25/4.03	ND	71.11/3.87	72.89/3.43	Cγ−Hγ in β−β (resinol) substructures (B)
Bα	84.45/4.64	ND	86.56/4.69	ND	Cα−Hα in β−β (resinol) substructures (B)
B’α	83.15/4.83	ND	82.55/4.91	ND	Cα−Hα in β−β (B’,tetrahydrofuran)
Aα	71.03/4.75	ND	71.60/4.86	ND	Cα−Hα in β–O–4 substructures linked to a G (A)
Aβ(S)	85.90/4.11	ND	86.02/4.11	ND	Cβ−Hβ in β–O–4 substructures linked to a S (A) (Erythro)
Aβ(S)	86.70/3.99	ND	86.57/3.99	ND	Cβ−Hβ in β–O–4 substructures linked to a S (A) (Thero)
Aβ(G/H)	83.55/4.30	ND	83.58/4.38	ND	Cβ−Hβ in β–O–4 substructures linked to a G/H (A)
X2	72.62/3.04	ND	72.70/3.05	ND	C2−H2 in β–D–xylopyranoside (X)
X3	73.62/3.22	ND	73.68/3.26	ND	C3−H3 in β–D–xylopyranoside (X)
X4	75.43/3.61	ND	75.25/3.52	76.18/3.17	C4−H4 in β–D–xylopyranoside (X)
X5	62.88/3.27	ND	62.86/3.18	ND	C5−H5 in β–D–xylopyranoside (X)
A′β(G)	80.82/4.54	ND	83.67/4.67	ND	Cβ−Hβ in acylated β–O–4 linked to a G unit (A)
Lα	ND	81.18/4.92	ND	81.37/4.90	Cα−Hα in lignocresol (L)
ICα	86.91/5.46	86.11/5.43	86.91/5.44	86.23/5.41	Cα−Hα in phenylcoumaran substructures (IC)
T′2,6	103.77/7.27	103.73/7.24	103.92/7.30	103.91/7.32	C′2,6-H′2,6 in tricin(T)
T6	98.61/6.24	98.64/6.28	98.75/6.23	98.75/6.21	C6-H6 in tricin(T)
T8	94.09/6.58	94.11/6.60	94.09/6.58	94.08/6.56	C8-H8 in tricin(T)
T3	105.20/7.04	105.23/7.04	104.50/7.02	104.51/7.02	C3-H3 in tricin(T)
G2	110.78/6.99	113.21/6.91	110.91/6.98	113.12/6.90	C2−H2 in guaiacyl units (G)
G′2	113.12/7.46	ND	111.50/7.53	ND	C2−H2 in guaiacyl units (G)
G5	114.79/6.77	114.83/6.65	115.52/6.78	114.73/6.62	C5−H5 in guaiacyl units (G)
G6	118.53/6.86	120.39/6.74	118.91/6.77	119.33/6.52	C6−H6 in guaiacyl units (G)
S2, 6	103.5/6.70	105.9/6.57	103.5/6.71	106.1/6.57	C2, 6−H2, 6 in syringyl units (S)
S′2, 6	106.2/7.18	106.5/6.85	106.3/7.33	110.8/7.38	C2, 6−H2, 6, C(α)=O in syringyl units (S′)
H3, 5	ND	116.8/6.66	ND	118.4/6.77	C3, 5−H3, 5 in p-hydroxyphenyl units (H)
H2, 6	127.7/7.18	127.2/7.26	127.9/7.20	125.1/7.07	C2, 6−H2, 6 in p-hydroxyphenyl units (H)
L2, 6	ND	129.7/7.10	ND	129.5/7.16	C2, 6−H2, 6 in lignocresol (L)
*p*CA2, 6	131.1/7.65	131.2/7.68	130.0/7.45	129.6/7.54	C2, 6−H2, 6 in *p*-coumarate (pCA)
*p*CA3, 5	ND	117.1/6.76	ND	116.1/7.00	C3, 5−H3, 5 in *p*-coumarate (pCA)
FA2	111.5/7.30	113.6/7.13	110.9/7.33	113.6/7.11	C2−H2 in ferulate (FA)
FA6	122.9/7.20	ND	122.9/7.13	120.4/6.65	C6−H6 in ferulate (FA)
FA8	ND	116.7/6.52	ND	ND	C8−H8 in ferulate (FA)
Fɑ	ND	129.4/6.95	ND	129.1/6.94	Cα−Hα in *p*-hydroxycinnamyl alcohol (F)
Fβ	ND	126.8/6.75	ND	126.8/6.74	Cβ−Hβ in *p*-hydroxycinnamyl alcohol (F)

δC/δH (ppm) a, HSQC attribution of the map of AHMWL sample. δC/δH (ppm) b, HSQC attribution of the map of AHLC sample. δC/δH (ppm) c, HSQC attribution of the map of RSMWL sample. δC/δH (ppm) d, HSQC attribution of the map of RSLC sample. ND, not detected.

**Table 9 molecules-28-01513-t009:** Carbohydrate compositions in the aqueous flour after phase separation.

Carbohydrate Composition	Areca Nut Husk	Rice Straw
Glucose (%)	34.15	41.92
Xylose (%)	17.84	13.97
Galactose (%)	0.92	1.34
Arabinose (%)	1.29	2.68
Mannose (%)	0.31	0.07
Total carbohydrate content (%)	54.51	59.98

## Data Availability

Data are contained within the article.

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
