# Peer review of "Study on Dissociation and Chemical Structural Characteristics of Areca Nut Husk"

_molecules, 2023, doi:10.3390/molecules28031513_

Round 1

Reviewer 1 Report

This paper analyzed the chemical composition and the lignin structure of areca nut husk. Detail information about biomass composition and structure can help to improve its utilization. This paper need to be improved before publication.

1.     The title of this paper should be revised since this manuscript didn’t study the depolymerization of areca nut husk.

2.     Abstract should be modified instead of just listing the results. 

3.     “ fiber morphologies, basic chemical compositions, lignin structures, and carbohydrate contens and compositions”, this statement should be rewritten accurately, especially for “basic chemical compositions” and “carbohydrate contents and compositions”.

4.     Why did the authors do the comparation study between areca nut husk and rice straw?  It should be stated briefly in the Introduction section.

5.     Wha’t the mainly goal and innovation of this study? Introduction section should be modified to declar the research objective of this study.

6.     Lines 117-119: I don’t understand the sentence “The contents of compounds…” what is the other thing beside “compounds”?  What’s the evidence for the smoother surface of areca nut husk?

7.     Lines 126-129: it is hard to say that areca nut husk is loosely organized based on the chemical composition results. In addition, the content of 1% NaOH extractive is quite difference between areca nut husk and rice straw.

8.     Both lignin and hemicellulose can be dissolved in alkaline solution. Why the content of NaOH extractive in rice straw (52%) is larger than the total content of lignin and hemicellulose of rice straw? (Table 1) what are the other alkaline-dissolved compounds?

9.     Lines 144-146, it is a little confused because rice straw is not a good potential material for pulp and paper making. Since authors compared between areca nut husk and rice straw, it is suggested to mention the application area which both of them could be served as the good potential raw materials.

10.   Lines 188-191, why the higher isolation yield can lead to the accurately of structural and quantification analysis?  In addition, the molecula weight of LC is smaller than that of MWL, which means that that the structural integrity of LC is poorer than that of MWL. Therefore, this statement is contradictory with other results.

11.   The figure captions of Figs 10 and 11 should be revised.

12.   Section 2.9, does it mean that almost all the carbohydrates can be be dissolved into the aqueous phase? What’s the yield of solid residue? Is it the same of Klason lignin? Authors used cresol to extract the carbohydrates during the acid hydrolysis process, and analysis it by GC. So what’s the distribution ratio of carbohydrates in this biphasic system? Can the GC results reflect the dissolation of carbohydrates? Also,it is suggested to modify the subheading of 3.13 since it is about the analysis of cresol phase.

13.   Conclusions: Key findings should be highly emphasized

14.   The language needs to be improved greatly .The whole manuscript shoud be checked carefully before publication. Many headlines, figure and table captions should be revised correctly. Following are some examples but not limited to these:

Lines 10-11, simplify the expression since there are many words with similar/same meanings.

Line 19: …higher, higher, and lower…

Line 53: …it is widely…grows

Line 56:…harvested from…

Line 56: Chinese province of Hainan

Line 59: …can grow per hectare

Line 125: 21.33 and….and …and…

Lines 152-154: the higher… strength of paper.

Lines 168-172: because MWL…rice straw.

Lines 252-254

Headline of 3.4

Reviewer 2 Report

This article is devoted to the study of depolymerization and chemical structural characteristics of areca husks. The article is written in a clear and accessible language. The main ideas are not in doubt. The abundance of various methods of analysis is a significant plus for this article and improves its quality. There are some points that I recommend to finalize:

1. Abstract is too exaggerated. You can shorten it, leaving the main one.

2. When describing the results of the study, it is desirable to use references to literary sources, to make more comparisons. This will give more solidity to some conclusions. This is important for all points of the article.

3. One can cite works on the study and depolymerization of lignin: 10.1007/s13399-020-00897-6, 10.1007/s13399-021-01637-0, 10.3390/catal11040467 and others.

4. It is desirable to make a stronger justification for the use of these particular plant materials for processing. Since they are partly characteristic of a particular region. Why is it relevant?

5. Conclusions can also be made more concise. They can be shortened and restructured.

Round 2

Reviewer 1 Report

The authors have answered all the questions carefully. I think it can be accepted for publication now.